# Bicoid gradient formation and function in the Drosophila pre-syncytial blastoderm

Zehra Ali-Murthy, Thomas B Kornberg*

Cardiovascular Research Institute, University of California, San Francisco, San Francisco, United States

**Abstract** Bicoid (Bcd) protein distributes in a concentration gradient that organizes the anterior/posterior axis of the Drosophila embryo. It has been understood that *bcd* RNA is sequestered at the anterior pole during oogenesis, is not translated until fertilization, and produces a protein gradient that functions in the syncytial blastoderm after 9–10 nuclear divisions. However, technical issues limited the sensitivity of analysis of pre-syncytial blastoderm embryos and precluded studies of oocytes after stage 13. We developed methods to analyze stage 14 oocytes and pre-syncytial blastoderm embryos, and found that stage 14 oocytes make Bcd protein, that *bcd* RNA and Bcd protein distribute in matching concentration gradients in the interior of nuclear cycle 2–6 embryos, and that Bcd regulation of target gene expression is apparent at nuclear cycle 7, two cycles prior to syncytial blastoderm. We discuss the implications for the generation and function of the Bcd gradient.

*For correspondence:
tkornberg@ucsf.edu

**Competing interests:** The author declares that no competing interests exist.

## Introduction

The discovery of the concentration gradient of Bicoid (Bcd) protein in the early Drosophila embryo established the existence and functional importance of a morphogen gradient for the first time (*Driever and Nusslein-Volhard, 1988a*, *1988b*; *Frigerio et al., 1986*); it was a watershed moment in developmental biology. These and subsequent studies showed that Bcd protein is present at the cortex of pre-cellular, syncytial blastoderm embryos, with levels that are highest at the anterior end and that decline exponentially toward the posterior (*Driever and Nusslein-Volhard, 1988b*; *Gregor et al., 2007*; *Spirov et al., 2009*). Although *bcd* mRNA is concentrated in the anterior cytoplasm of stage 13 oocytes and of embryos immediately after egg laying (*Berleth et al., 1988*; *Frigerio et al., 1986*; *Riechmann and Ephrussi, 2004*), its distribution extends more posteriorly in the embryo at syncytial blastoderm stages (*Berleth et al., 1988*; *Frigerio et al., 1986*; *Spirov et al., 2009*). Whether the protein gradient forms by passive diffusion following synthesis of Bcd protein at more anterior locations (*Gregor et al., 2007*; *Little et al., 2011*), or is produced in place by the *bcd* mRNA concentration gradient is in dispute (*Fahmy et al., 2014*; *Spirov et al., 2009*).

After fertilization, nuclei divide rapidly and synchronously eight times in the interior of the embryo, moving outward in a choreographed sequence that places them simultaneously at the surface at nuclear cycle 9 (nc9). The five division cycles that follow delineate the syncytial blastoderm stages nc10-nc14. Nuclear divisions cease at nc14, whereupon the nuclei begin to individuate into single cells and gastrulation ensues. Various measures, including in situ hybridization (*Erickson and Cline, 1993*; *Pritchard and Schubiger, 1996*), RT-PCR (*Harrison et al., 2010*), genome array hybridization (*De Renzis et al., 2007*; *Little et al., 2011*; *Lu et al., 2009*), RNA seq (*Lott et al., 2011*), DNA footprinting (*Harrison et al., 2010*), chromatin profiling (*Harrison et al., 2011*) and ChIP-seq (*Blythe and Wieschaus, 2015*), show that the zygotic genome is transcriptionally activated during the syncytial blastoderm period.

**eLife digest** As an embryo develops, a single cell transforms into a collection of different types of cells. One protein that is crucial for this process in fruit fly embryos is Bicoid. Thirty years ago, scientists discovered that Bicoid protein is concentrated at the head end of the embryo and gradually decreases in amount towards the rear end. This concentration gradient of Bicoid protein organizes the embryo body and regulates the expression of many genes, thus directing the cells to develop different identities.

Several assumptions had been made about how this gradient is established. It was thought that in the unfertilized egg, the mRNA molecules that will be translated to produce Bicoid proteins are stored in an inactive state in the region of the egg that later develops into the embryo's head. In the embryo, the mRNA molecules were believed to remain in the head region while being translated, with the newly formed proteins then gradually spreading from this site to create the Bicoid gradient. It was also thought that no Bicoid proteins are stored in the unfertilized egg. However, no known methods were sensitive enough to investigate these assumptions.

Now, using newer and more sensitive methods, Ali-Murthy and Kornberg show that Bicoid protein is present in the unfertilized fruit fly egg in the same region as the mRNA molecules that make Bicoid. Furthermore, the Bicoid gradient forms when the embryo has fewer than 32 nuclei, much earlier in development than previously thought. The Bicoid protein also does not appear to spread passively towards the rear of the embryo, but is transported in a more orchestrated manner.

Overall, Ali-Murthy and Kornberg's results suggest that the early fruit fly embryo is more organized and actively regulated than had been previously understood. This paves the way for further studies that use sensitive techniques to investigate this early stage of development.

Oogenesis provides the Drosophila egg with a rich dowry of mRNA that is essential to the development of the early, pre-cellular embryo, and for a number of reasons, the period that precedes the maternal-to-zygotic transition has been considered to depend only on maternal stores and to be independent of the zygotic genome. One, the early nuclear divisions are so rapid (9.6 min) that productive gene expression has been deemed impossible. Two, molecular analyses of transcriptional activity have almost universally failed to detect RNA synthesis at pre-syncytial blastoderm stages, even as the sensitivity of the detection methods has increased. Three, comprehensive genetic screens for mutants defective in early development identified many genes that are required maternally, but found no evidence for genes that must be active in the zygote prior to cellularization at nc14 (*Luschnig et al., 2004*; *Merrill et al., 1988*; *Perrimon et al., 1984*; *Schupbach and Wieschaus, 1989*, *1991*; *1986*).

Although these observations have substantiated the idea that the gene products supplied by the mother during oogenesis are sufficient for first thirteen cleavage cycles, this conclusion is based on negative findings, and because it depends on the sensitivity of the analysis, it leaves open the possibility that more sensitive methods might detect zygotic transcripts expressed from a small number of active genes or might recognize phenotypes in mutant embryos that were not revealed by then available histological techniques. Drosophila embryos are heavily populated with yolk and glycogen granules that impede histological studies, and have few obvious morphological features that can be evaluated for dependence on genotype. In addition, the idea that rapidly dividing nuclei are incapable of expression has no experimental basis because the capacity for transcription and translation at early nuclear cycles has not been analyzed. It is possible therefore that the normal transcriptional processes are sufficient for transcription units that are small (approximately 70% of transcripts made by nc10-12 embryos lack introns; *De Renzis et al., 2007*), or it may be that yet unexplored mechanisms produce and use transcripts more rapidly at early stages.

There are, in fact, several reports of expression by the zygotic genome in pre-syncytial blastoderm Drosophila embryos. The earliest reported zygotic expression obtained by in situ hybridization is at nc8 for the gene *sis-A* (*Erickson and Cline, 1993*). Evidence for earlier gene expression (β-galactosidase activity in nc4 embryos) was reported for a transgene construct that carried the *lacZ* gene regulated by a FTZ-F2 enhancer fragment (*Brown et al., 1991*). The strongest evidence for

functional expression in pre-syncytial blastoderm embryos is for the *engrailed (en)* gene (*Ali-Murthy et al., 2013*; *Karr et al., 1985*). En protein synthesized in the embryo was detected by antibody staining in nc2 nuclei, a mutant phenotype was identified in *en* nc2-3 embryos, and PCR and RNA-seq studies provided evidence for expression of a small cohort of genes prior to nc7 (*Ali-Murthy et al., 2013*). These findings show that development of the early embryo is not entirely pre-programmed and that the processes that orchestrate the early stages are actively and directly regulated.

The work described here investigates the development of the early embryo further, focusing on the formation and function of the Bcd gradient. To do so, we needed to develop methods that overcome the technical impediments that have limited analysis of the early stages. For instance, because Drosophila fertilization cannot be made synchronous and because females hold their embryos for variable periods prior to laying, an embryo of any particular stage must be identified histologically and individually, and methods must be used that are sensitive enough to detect expression from a small number of active genes in the small number of nuclei. Although molecular detection methods have improved, antibody staining and in situ hybridization have been ineffective due to the large number of yolk and glycogen granules that reduce signal to noise and sensitivity. By using sensitive methods that we modified for studies of the early embryo, we found that the processes that generate the Bcd protein gradient are more complex and operate earlier than had been appreciated, and that the function of the Bcd gradient begins prior to formation of the syncytial blastoderm.

## Results

### Bicoid-dependent Krüppel expression

*Krüppel (Kr)* is a 'gap'" gene that is expressed by pre-cellular, syncytial blastoderm embryos in a central band that spans approximately 20% embryo length. Although the earliest reported expression detected by in situ hybridization is nc10 (*Pritchard and Schubiger, 1996*), RNA-seq and QT-PCR that we carried out previously detected *Kr* transcripts prior to nc10 in pre-syncytial blastoderm embryos (*Ali-Murthy et al., 2013*). To identify the nuclei that express *Kr* transcription prior to nc10, we evaluated various different techniques for in situ hybridization, and determined that a procedure that uses DIG probes to be the most sensitive for our studies of *Kr* and *bcd* transcripts in pre-cellular embryos (see Materials and Methods and *Figure 1—figure supplement 1*).

*Figure 1* shows images of nc7-13 embryos in which *Kr* transcripts were detected. In situ hybridization can detect sites of nascent transcript production as points of staining or fluorescence (*Femino et al., 1998*; *Shermoen and O'Farrell, 1991*), and the nc10-13 embryos we analyzed had bright dots in most or all nuclei in the 'Kr band'. The fraction of nuclei with dots was lower in nc7-9 embryos, but most of the nuclei with dots were in the central region where the Kr band forms. The position of the Kr band, but not the number of Kr dots, was sensitive to maternal *bcd* gene dosage: in nc12 embryos, it was at 43–64% embryo length (from anterior end) if mothers were wild type (had two *bcd* copies) and 54–75% if mothers had six. This is indicative of the dependence of *Kr* expression on maternal Bcd (*Figure 1—figure supplement 2*), and is consistent with earlier studies (*Driever and Nusslein-Volhard, 1988a*; *Hoch et al., 1991*).

The images in panels in *Figure 2* show nuclear dots in nc9 and nc11 embryos. The dots were absent from early interphase nuclei, but increased in number and brightness as the cycle progressed, and they reached peak brightness at late interphase-early prophase (*Figure 2A,B*). The level of fluorescence in the cytoplasm also increased during the nuclear cycles. Both nuclear dot and cytoplasmic fluorescence was higher at nc11 relative to nc9, and was also higher in embryos produced by mothers with six *bcd* genes. The panel with a high magnification view of a nc9 pro-metaphase nucleus shows that both dots in this nucleus appear to be at visible chromosome arms (*Figure 2C*). Nuclei in embryos that have one *Kr* gene had only one dot (*Figure 2D*). These observations suggest that the in situ hybridization method detected nascent transcripts that were chromosome-associated, as well as mature transcripts in the cytoplasm. The increase in dot brightness during the nuclear cycle suggests that the in situ hybridization signal is proportional to the quantity (number and length) of transcripts, and although the longer nuclear cycle at nc11 relative to nc9 may account for the increased signal at the later cycle, the signal increase observed in embryos with additional Bcd dosage was not expected.

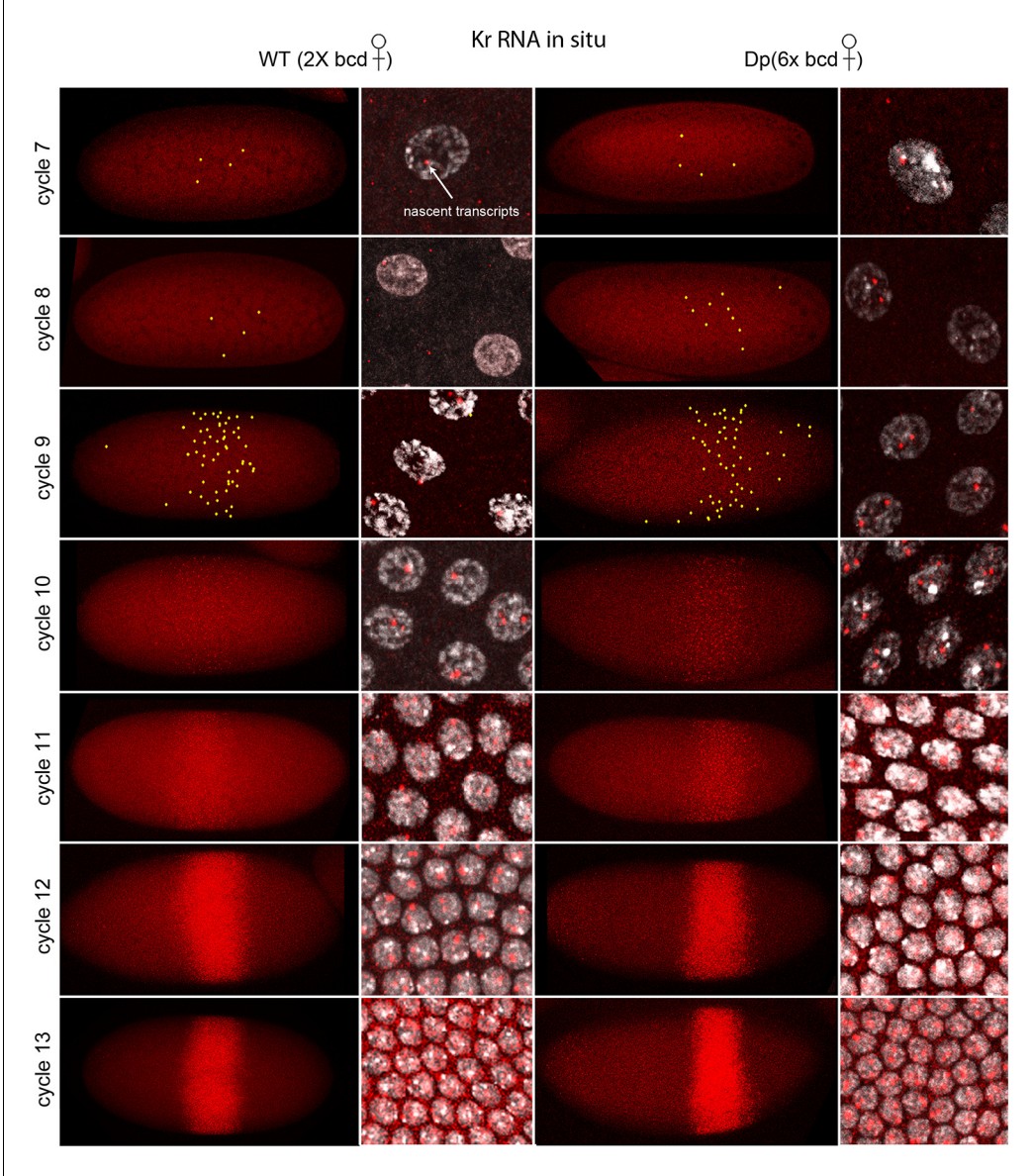

**Figure 1.** In situ detection of *Kr* transcipts in pre-cellular embryos. In situ hybridization detected *Kr* RNA (red) in nuclear cycle 7–13 embryos that were produced by females with two (left two columns) and six (right two columns) *bcd* gene copies. Embryos are prophase stage, dorsal up and anterior left. Embryo images are projection composites from serial optical sections; high magnification images show nuclei with red fluorescent dots. The number of nuclei with red dots and the dot brightness increases with each successive cycle. Two dots are visible in most nuclei; some have only one but none have three. Yellow dots in nuclear cycle 7–9 embryos indicate dots that were too faint to be visible at low magnification. The width of the band of *Kr*-expressing nuclei was approximately the same in the two genotypes; its position shifted more posteriorly in the embryos with excess Bcd.

The following figure supplements are available for figure 1:

**Figure supplement 1.** Methods of in situ hybridization compared.

**Figure supplement 2.** Dependence of gastrula morphology on maternal genotype.

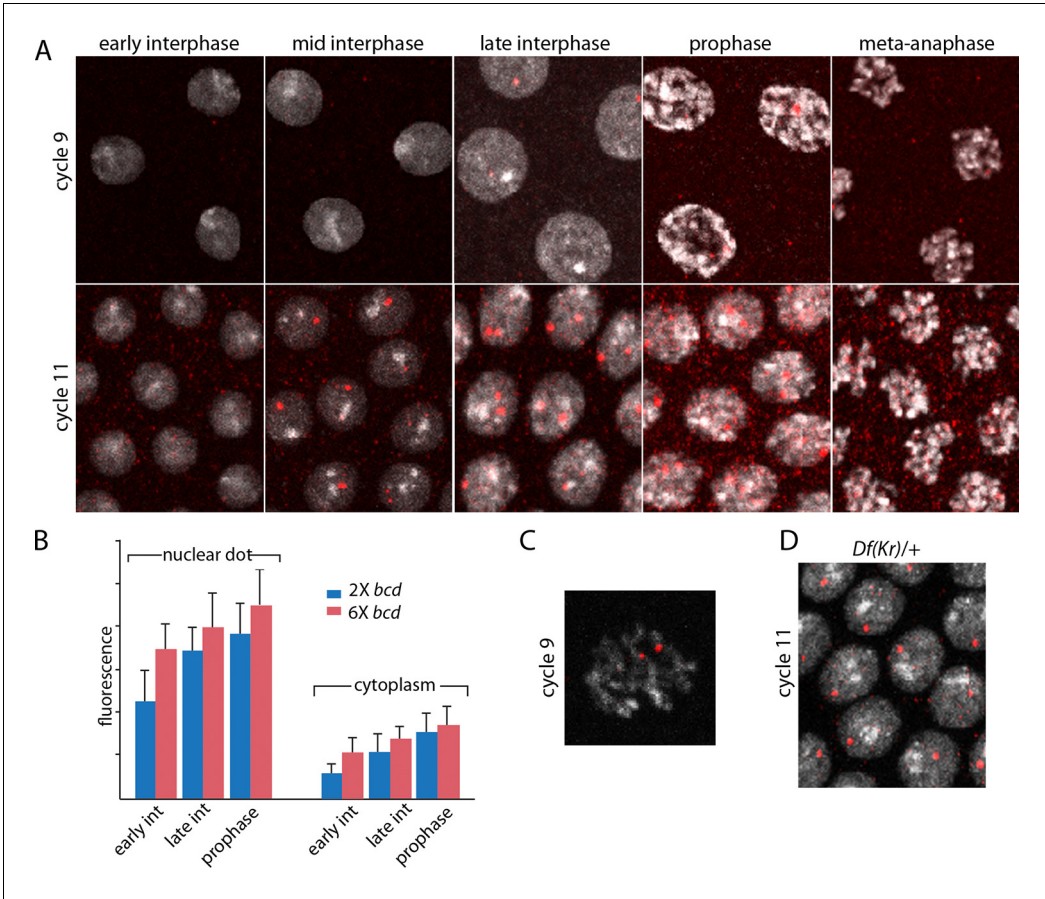

**Figure 2.** Characterization of *Kr* transcripts detected in situ. (**A**) In situ hybridization detected *Kr* transcripts in mid-interphase to late prophase nuclei as discrete red fluorescent dots in nuclear cycle 9 and 11 embryos. (**B**) Bar graphs depicting measures of red fluorescence nuclear dots in nc11 nuclei and in a boxed area of adjacent cytoplasm quantify the dependence on nuclear cycle stage and Bcd level in embryos from mothers with two and six *bcd* genes. n=10 for nuclear dots, n=4 for cytoplasm. Differences between 2x and 6x are significant for nuclear dots (two-tailed value <0.0001 and t-test <E-08) and for early interphase cytoplasm (two-tailed value <0.0001 and t-test 3E-07). (**C**) High magnification image of a cycle 9 pro-metaphase nucleus revealing the apparent association of two red fluorescent dots with chromosome arms. (**D**) Nuclear cycle 11 nuclei in *Kr/+* embryos have only one red fluorescent dot.

The following source data is available for figure 2:

**Source data 1.** Source data for 2B.

The number of nuclei with fluorescent dots at nc7 and nc8 was small, and although the position of these marked nuclei is in the central region of the embryo where the Kr band will be robust at later cycles, it is important to know whether the dots in these nuclei mark nuclear transcripts. Chromosomes in nuclei at the cortex of syncytial blastoderm embryos are organized and polarized such that the centromeric regions congregate apically and the telomeres congregate basally (*Marshall et al., 1996*). Chromosome organization in nuclei at earlier, pre-syncytial blastoderm stages has not been examined previously. *Figure 3* characterizes nuclear and background dots in nc7 and nc8 embryos with respect to proximity to the embryo cortex. Nuclei at nc7 and nc8 are approximately 34 μm and 22 μm, respectively, from the embryo surface (*Figure 3 C,D*). Comparison of fluorescence levels of these dots in successive confocal optical sections revealed that whereas the dots in nuclei were in the basal sections, dots not in nuclei (and attributed operationally to background) were apical. Because the *Kr* gene is at the tip of 2R (polytene band 60F5), this result

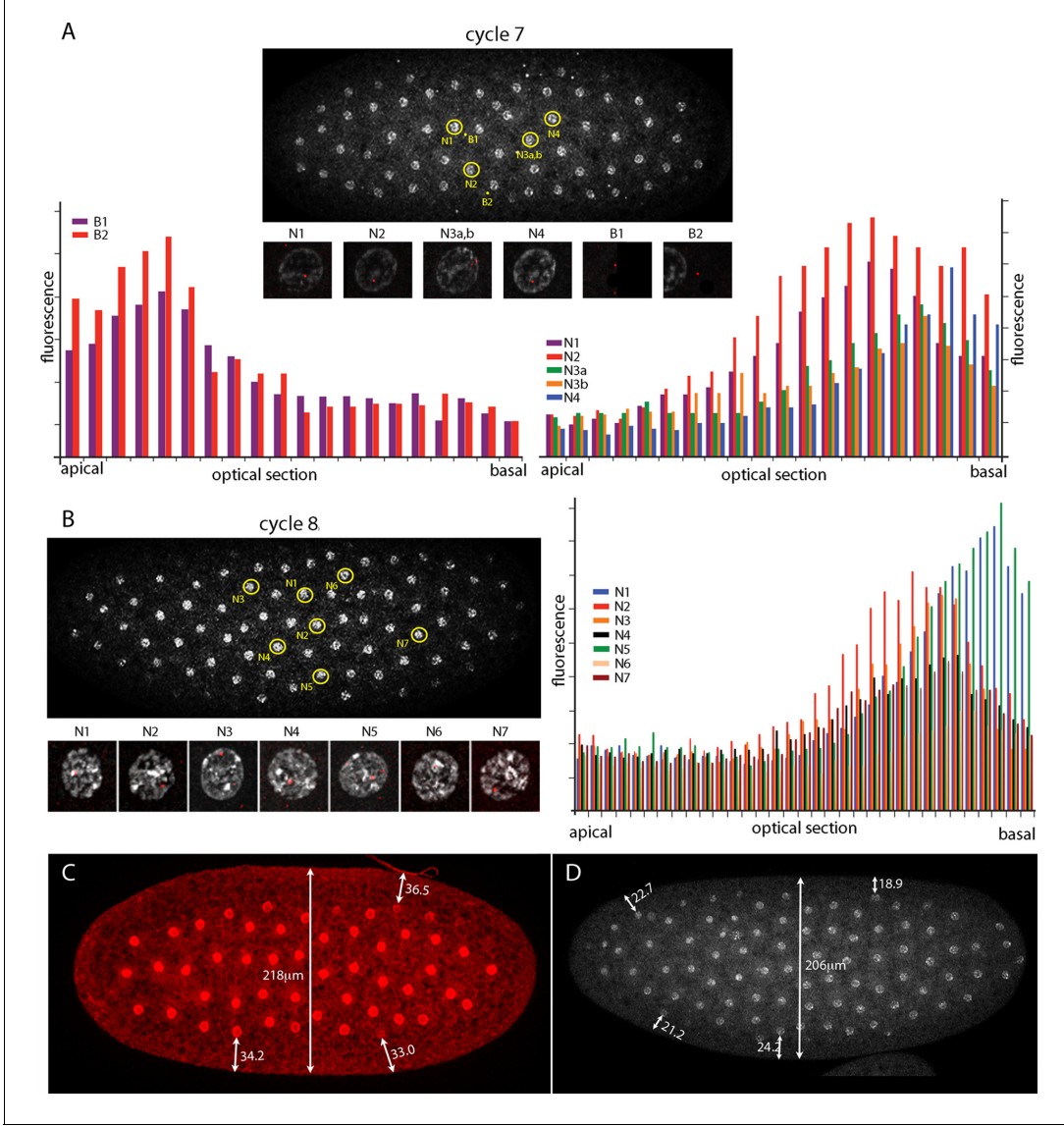

**Figure 3.** Intranuclear localization of *Kr* transcripts in pre-syncytial blastoderm nuclei. (**A,B**) In situ hybridization detected *Kr* transcripts (red fluorescence) in nuclear cycle 7 and 8 embryos; prophase, dorsal up, anterior left. Yellow circles indicate dots whose red fluorescence was measured in successive optical planes and that are depicted in the bar graphs. (**A**) Two fluorescent non-nuclear dots (B1, B2) that were near nuclei with red dots were analyzed in the nuclear cycle 7 embryo (left graph); four fluorescent dots (N1-4) were analyzed (right graph). (**B**) Seven red nuclear dots were analyzed in the nuclear cycle 8 embryo. (**C,D**) Nuclei imaged with anti-nuclear lamin (**C**) and DAPI (**D**) show the placement of nuclei at nuclear cycle 7 (**C**) and nuclear cycle 8 (**D**) and the measured distances (μm) between nuclei and the cortex.

The following source data is available for figure 3:

**Source data 1.** Source data for 3A,B.

suggests that nuclei are polarized prior to their arrival at the cortex and it is consistent with the conclusion that the nuclear dots mark sites of *Kr* transcription.

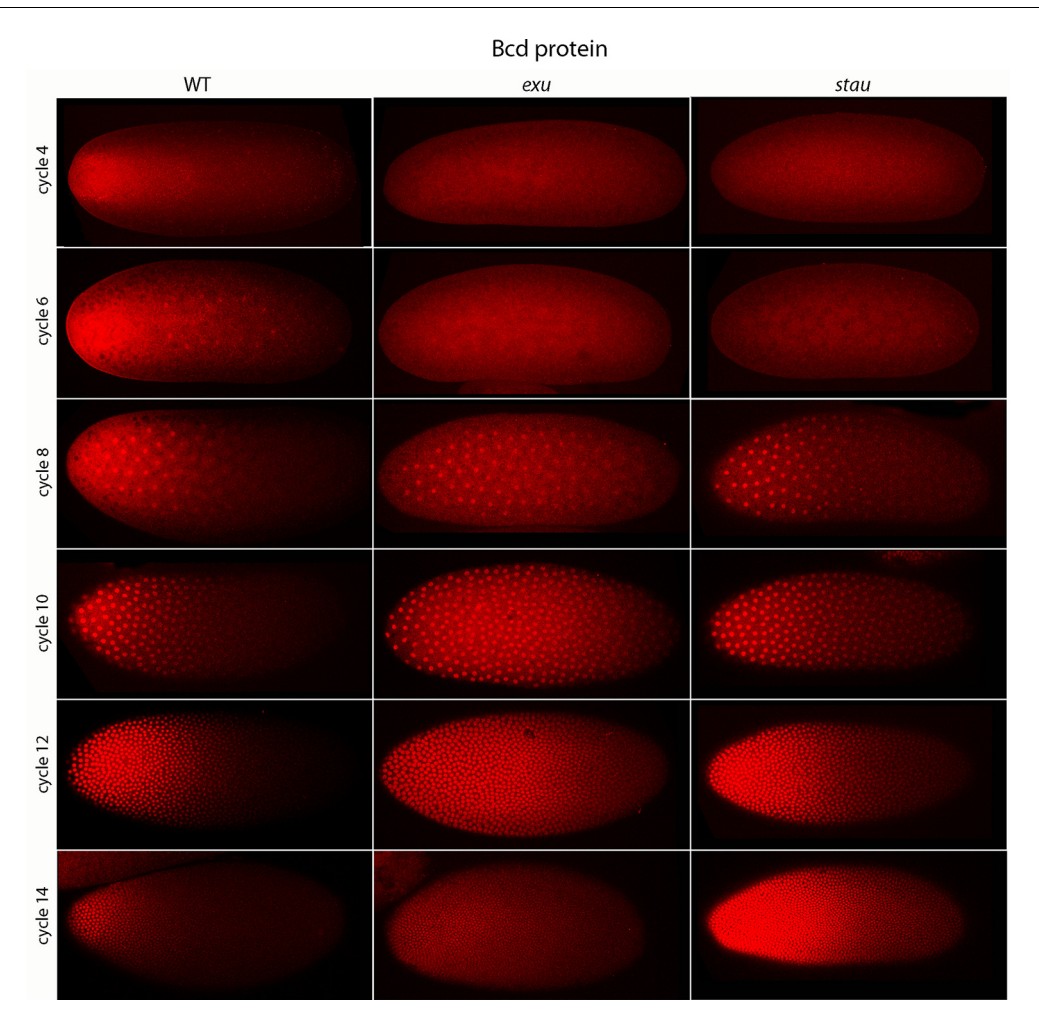

**Figure 4.** Bcd protein distributions in pre-cellular embryos. Anti-Bcd antibody detected Bcd protein (red fluorescence) in nuclear cycle 4–14 WT, *exu* and *stau* embryos. Embryos are prophase, dorsal up and anterior left. Images are projection composites from serial optical sections (46–50 per embryo, spanning from the dorsal to ventral surface). Fluorescence is brightest at the anterior end and in the mutant embryos it is brighter and extends more posteriorly than in the WT. Fluorescence is apparent concentrated in nuclei of nuclear cycle 6–14 embryos.

The following source data and figure supplements are available for figure 4:

**Source data 1.** Source data for *Figure 4—figure supplement 1*.

**Figure supplement 1.** *bcd* RNA levels in pre-cellular WT and *exu* and *stau* mutant embryos.

**Figure supplement 2.** *Kr* expression in *exu* and *stau* mutant embryos.

## *Bicoid* expression in pre-cellular embryos

Previous studies described the Bcd concentration gradient at the syncytial blastoderm stages (*Driever and Nusslein-Volhard, 1988b*; *Fahmy et al., 2014*; *Little et al., 2011*; *Spirov et al., 2009*), but the earlier, pre-syncytial blastoderm stages have not been characterized. We developed a method for antibody staining of early embryos that has high signal-to-noise sensitivity (see Materials and methods), and used it to monitor Bcd in the pre-cellular stages. *Figure 4* shows dorsal views of nc4-14 embryos that were either from normal mothers (WT), or that were from mothers that were mutant for *exuperentia (exu)* or *staufen (stau)*. Function of the *exu* and *stau* genes during oogenesis is essential to localize *bcd* mRNA at the anterior pole and for the Bcd gradient in syncytial

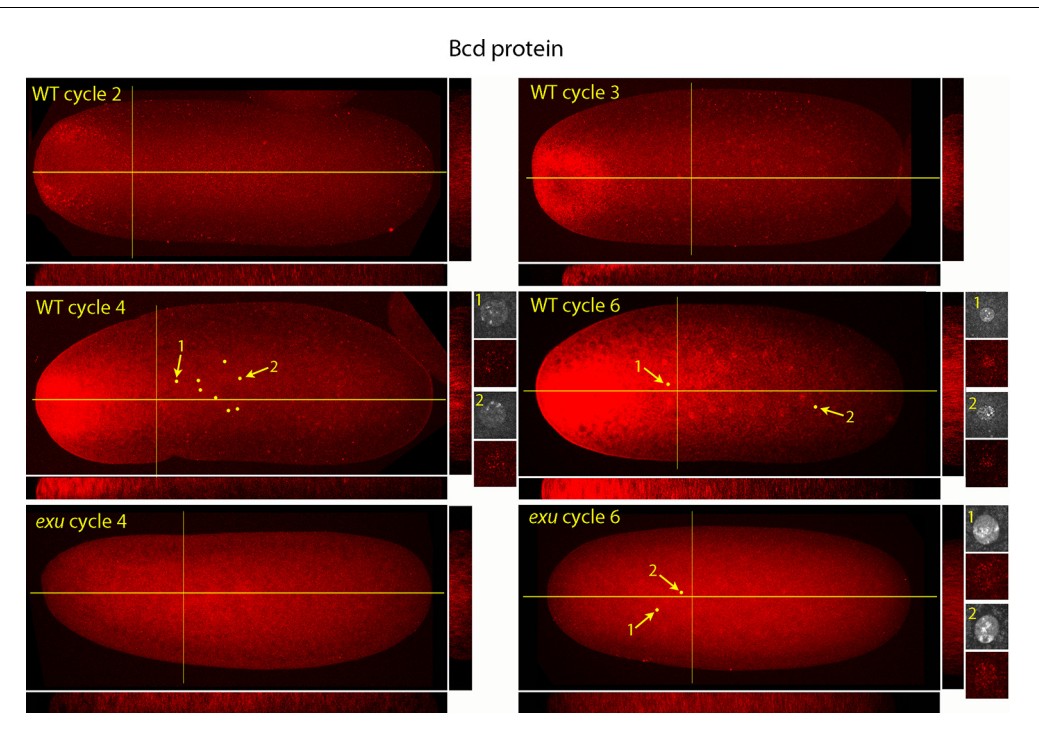

**Figure 5.** Bcd protein distributions in nuclear cycle 2 to 6 embryos. Anti-Bcd antibody detected Bcd protein (red fluorescence) in nuclear cycle 2 to 6 WT and *exu* embryos. Embryos are prophase, dorsal up and anterior left. Images are projection composites from serial optical sections. Yellow lines indicate the position of mid embryo sagittal and transverse optical sections that are shown below and to the right, respectively, of each embryo. The anterior to posterior fluorescence gradient is apparent in the sagittal sections of WT but not the *exu* embryos; internal embryo fluorescence is apparent in the transverse sections of both WT and mutant embryos. Nuclei in the WT nuclear cycle 4 (indicated by yellow dots), nuclear cycle 6 and *exu* nuclear cycle 6 embryos concentrate red fluorescence; high magnification images of the indicated nuclei show DAPI fluorescence and resolve spots of red fluorescence in both WT and *exu* nuclei.

blastoderm embryos (reviewed in *Kugler and Lasko, 2009*). The Bcd protein gradient detected by this method in WT and mutant embryos at syncytial blastoderm stages was consistent with previous reports. It extended to approximately 60% embryo length in WT nc12 embryos. In the syncytial blastoderm stages of embryos from mutant mothers, the gradient extended more posteriorly and the apparent levels of Bcd protein were higher than WT, especially at late nc14 when Bcd protein was low in WT. Although the Bcd protein levels appeared to be higher in the mutant embryos, Q-PCR analysis showed that the levels of *bcd* RNA was similar in the WT and mutants (*Figure 4—figure supplement 1*). The late persistence of Bcd protein was particularly pronounced in embryos that lacked maternal *stau*. *Kr* expression in *exu* and *stau* embryos was robust but misplaced, and was broader than the normal in nc13 and nc14 *stau* embryos (*Figure 4—figure supplement 2*).

A concentration gradient of Bcd protein was also observed in nc4 and nc6 embryos that were produced by WT females. The apparent level of Bcd was higher in the nc6 embryos and the gradient extended more posteriorly. In contrast, staining by the anti-Bcd antibody did not detect a gradient distribution of Bcd in nc4 and nc6 embryos from *exu* or *stau* mothers (*Figure 4*, upper panels).

*Figure 5* shows a higher resolution analysis of Bcd distribution in pre-syncytial blastoderm embryos. The panels show projection images and include sagittal sections calculated from the stacks of optical sections. In the nc2 embryo, Bcd levels were low medially at the anterior end, high in the surrounding region and extended approximately 20% embryo length posteriorly. In the nc3 embryo, both the 'doughnut' distribution at the anterior pole and the posteriorly extending medial plume were more pronounced. The medial plume was increased in the nc4 embryo, and more so in the nc6 embryo. Nuclei in these embryos were in the interior and enveloped by the medial plume and had

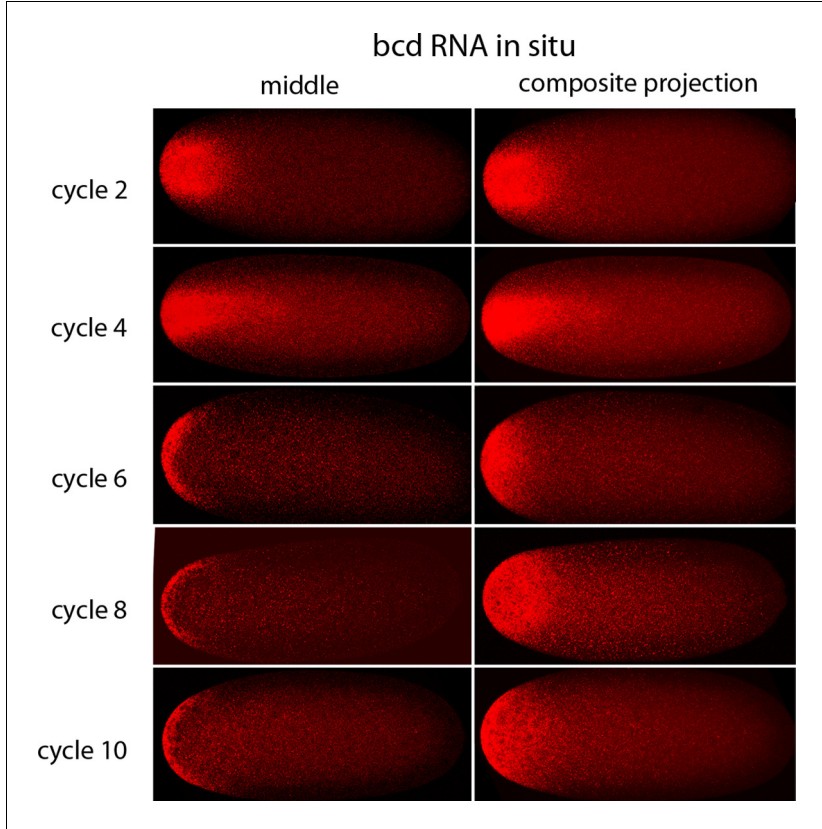

**Figure 6.** In situ detection of *bcd* transcripts in pre-cellular embryos. In situ hybridization detected *bcd* transcripts (red fluorescence) in nuclear cycle 2, 4, 6, 8 and 10 embryos. Embryos are dorsal up and anterior left. Images are mid-embryo optical sections (left) and projection composites from serial optical sections (right).

multiple fluorescent puncta. The nuclei appeared to concentrate Bcd protein, with higher levels of fluorescence in the more anterior nuclei. Fluorescence was apparent only in the most anterior nuclei of nc4 embryos but could be detected in a concentration gradient that encompasses all of the nuclei in nc6 embryos. Analysis of composite projection images and optical sections showed the Bcd protein detected by the antibody was mostly in the interior regions and increased in amount and extended more posteriorly from nc2-nc6. Although no medial plume was observed in nc4 and nc6 embryos that were from *exu* mothers, staining by the anti-Bcd antibody was apparent and fluorescent punctae were visible in nuclei of the nc6 mutant embryo. This indicates that Bcd protein was present in the interior of mutant pre-syncytial blastoderm embryos, although its distribution was apparently more uniform than WT.

## *bcd* RNA distributions in pre-cellular embryos

In situ hybridization detected *bcd* RNA concentrated at the anterior end of pre-cellular stages (*Figure 6*) in patterns that were similar to Bcd protein (*Figures 4,5*). *bcd* RNA was concentrated at the anterior end of nc2 embryos, and in nc4 embryos formed a plume that extended posteriorly in the embryo interior. It was predominantly anterior and cortical in embryos at stages nc8 and older. These images of *bcd* RNA in syncytial blastoderm stage embryos are consistent with previous reports (*Little et al., 2011*; *Spirov et al., 2009*). Higher resolution examination of the anatomy of the RNA distribution at nc1 and nc3 stages (*Figure 7*) revealed that the RNA was tightly restricted at the anterior end at nc1, but at nc3 was distributed in an internal plume that had distinct contours and dorsal/ventral asymmetries. The posterior extent of the plume was greatest at nc4, and was not as great at nc5 and nc6.

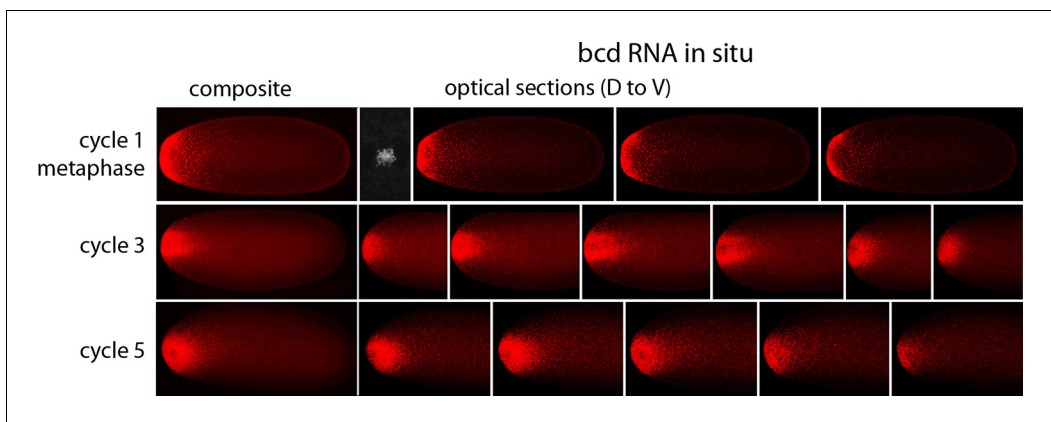

**Figure 7.** In situ detection of *bcd* transcripts in nuclear cycle 1, 3 and 5 embryos. In situ hybridization detected *bcd* transcripts (red fluorescence) in nuclear cycle 1, 3, and 5 embryos. Embryos are dorsal up and anterior left. Left panels are projection composites from serial optical sections and panels to their right are successive dorsal to ventral optical sections. DAPI-stained image of nucleus shown for the nuclear cycle 1 embryo.

## *bcd* expression in oocytes

Published in situ hybridization and antibody studies report the detection of *bcd* RNA and protein in embryos, but only *bcd* RNA in oocytes (*Berleth et al., 1988*; *Driever and Nusslein-Volhard, 1988b*; *Frigerio et al., 1986*; *Schnorrer et al., 2000*). Although Bcd protein was detected in unfertilized eggs, the assumption was made that stage 14 oocytes have anterior-localized RNA but no Bcd protein, and that the protein in unfertilized eggs was produced in conjunction with egg activation and egg laying. However, because there have not been suitable methods for fixation and preparation, previous studies of stage 14 oocytes have been limited to in situ hybridization analysis of ovarian sections (*Frigerio et al., 1986*). In contrast to younger oocytes, stage 14 oocytes have vitelline membrane and chorion layers that reduce permeability, but they are sensitive to hypochlorite and organic solvent treatments that are resisted by the fully formed vitelline membrane and chorion layers of subsequent stages. We sought methods to examine *bcd* RNA and protein in stage 14 oocytes because our analysis revealed that significant levels of Bcd protein are present in early nc1 embryos and unfertilized eggs (*Figure 8*). The question we addressed was whether the protein had been produced prior to fertilization or in the short interval between fertilization and nc1 metaphase.

At the beginning of stage 14 (stage 14a), follicle cells completely envelope the oocyte (*Figure 8*), the chorionic dorsal appendages are not yet separated, and no nurse cells remain. During stage 14, the oocyte 'undresses': the follicle cells unfasten from the chorion and re-arrange into longitudinal rows and the nuclei elongate (stage 14b), and the cells begin to migrate toward the anterior end and concentrate in an anterior 'collar' (stage 14c). The follicle cells are shed from the tips of the fully formed and separated dorsal appendages (stage 14d). At stage 14c-d, oocytes exit from the distal end of the ovary and enter the lateral branches of the oviduct. Staining with anti-Bcd antibody did not detect Bcd protein in stage 13 (not shown) or stage 14a–b oocytes (*Figure 8H,I*), but detected Bcd protein in stage 14c (not shown) and 14d oocytes (*Figure 8L*). Bcd protein was present in a tightly delineated band that follows the contours of the oblique concavity at the oocyte's anterior end. This concavity does not fully resolve into a blunt nose cone until mid nc1. In situ hybridization also detected *bcd* RNA in a band that follows the anterior concavity of stage 14 oocytes. This RNA distribution was similar to the Bcd protein except that in contrast to the protein, it was detected at stage 12 (*Figure 8—figure supplement 1*) and 13 (not shown) and all periods of stage 14. These results show that the translation block that inhibits Bcd protein production is relieved prior to fertilization and egg activation, and shows that Bcd protein is produced and is present precisely at the site where *bcd* RNA is sequestered.

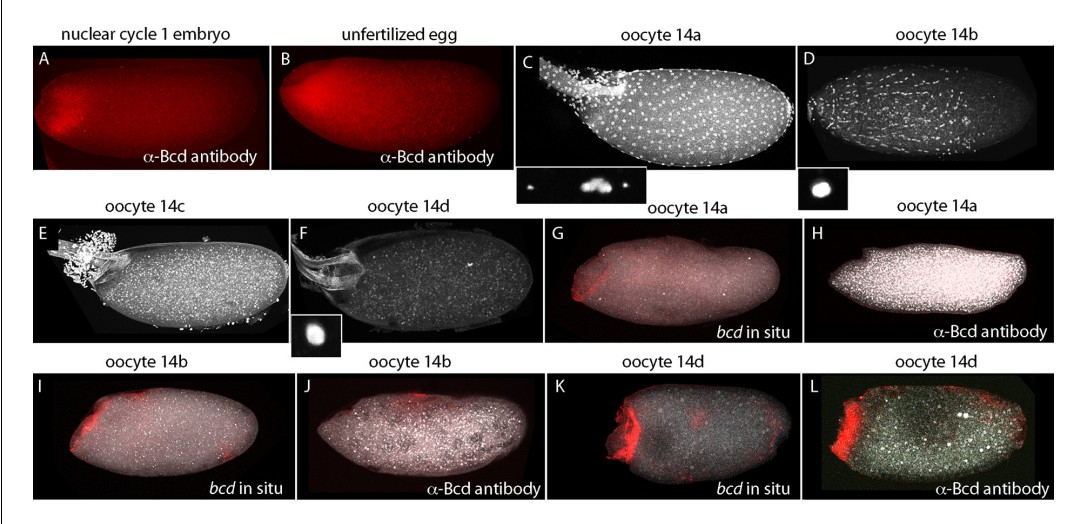

**Figure 8.** *bcd* expression in stage 14 oocytes. α-Bcd antibody detects Bcd protein in nuclear cycle 1 embryos (**A**) and in unfertilized eggs (**B**). Stage 14 oocytes (stained with DAPI) are covered with follicle cells initially and the dorsal appendages are juxtaposed (stage 14a; (**C**)); the nuclei of the follicle cells elongate and the follicle cells migrate anteriorly (stage 14b; (**D**)), form an anterior ring stage 14c; (**E**) that extrudes distally from the dorsal appendages, which separate from each other stage 14d; (**F**). High magnification images of DAPI-stained nuclei in (**C,D,F**) show the oocytes to be at prophase (14a) and true metaphase (14b, 14d) (*Gilliland et al., 2009*). In situ hybridization detects *bcd* RNA at all stage 14 oocytes (**G,I,K**); antibody staining detects Bcd protein in stage 14d oocytes (**L**) but not in younger stages (**H,J**). Orientation anterior left.

The following figure supplement is available for figure 8:

**Figure supplement 1.** In situ detection of bcd transcripts in stage 12 oocyte.

## Discussion

Our discovery that the zygotic genome is transcribed by nc2-8, pre-syncytial blastoderm embryos (*Ali-Murthy et al., 2013*) led us to investigate how patterns of *Kr* expression might change between nc1-14. *Kr* transcripts were identified by our Q-PCR analysis of nc3-6 embryos, and because *Kr* is expressed by cortical nuclei in a discrete band that straddles the middle region of nc10-14 syncytial blastoderm embryos (*Gaul and Jackle, 1987*; *Pritchard and Schubiger, 1996*), we sought to understand its expression pattern at earlier stages. Was *Kr* expression by nc3-6 embryos also spatially limited, or was it expressed by all nuclei in a uniform pattern? The improved method we developed to monitor RNA in situ did not detect *Kr* in nc3-6 embryos, presumably due to insufficient sensitivity, but the unexpected finding that *Kr* is expressed by nuclei in the middle region of embryos as early as nc7 has fascinating implications that this work explores.

### Nuclear polarity prior to syncytial blastoderm

The cortical nuclei at the surface of late syncytial blastoderm embryos have apical-basal polarity that is manifested in the organization of their chromosomes. Whereas the centromeric regions of the chromosomes coalesce to form a chromocenter that is at the most apical aspect of the nuclei, the telomeres congregate basally (*Marshall et al., 1996*). Our in situ hybridization experiments located nascent transcripts at the *Kr* gene, which is situated just proximal to the telomere of chromosome 2R, and analysis of serial optical sections revealed that the nascent transcripts are at the basal regions of nc7 and nc8 nuclei (*Figure 3*). These nuclei are approximately 34 µm and 22 µm, respectively, from the embryo surface, indicating that nuclei in the interior of the embryo are organized and polarized with respect to the cortex. The idea that internal chromosome organization is related to gene expression (*Marshall et al., 1996*) is consistent with our finding that these nuclei are transcriptionally active. Nuclear divisions in the pre-syncytial blastoderm embryo are precisely synchronized even as the nuclei move to collectively occupy larger volumes of the embryo at later nuclear cycles, and the choreography they follow as they move to the cortex is highly reproducible. Our

finding that nuclei are oriented prior to their arrival at the cortex indicates that their apical-basal polarity is not a consequence of short-range interactions with the cortex, and suggests that a global system of structural organization and regulation exists in the early embryo.

## Bicoid protein production

The Bcd concentration gradient is generated from mRNA that is made by nurse cells and is transported to the oocyte where it is sequestered in an inactive state at the anterior end (*Frigerio et al., 1986*). Previous studies reported that Bcd protein was absent from stage 13 oocytes but present in embryos and unfertilized eggs (*Driever and Nusslein-Volhard, 1988b*). Although stage 14 oocytes were not examined, the assumption that Bcd protein is also absent from stage 14 oocytes was the basis for the current model that translation of Bcd protein begins at egg activation. Our finding that Bcd protein is made in late stage oocytes prior to activation and fertilization (*Figure 8*) revises this model, but does not fundamentally change it. The distribution of Bcd protein in stage 14c and 14d oocytes is consistent with the idea that Bcd protein is made by mRNA that was localized to the anterior end. Its production in the oocyte raises the question whether Bcd protein is made only in the oocyte and not in the embryo, but this seems unlikely because the level of staining by α-Bcd antibody increased during the pre-cellular cycles (*Figure 4*).

## *bcd* RNA and Bcd protein distribution in the pre-syncytial blastoderm embryo

Two models have been proposed for the formation of the Bcd protein gradient. One is based on the observation that both *bcd* mRNA and Bcd protein distribute in concentration gradients at syncytial blastoderm stages (nc9-14), and because of the spatial relationship between these distributions, it posits that the Bcd protein gradient is a direct consequence of the location of *bcd* mRNA (*Fahmy et al., 2014*; *Spirov et al., 2009*). This model does not require or invoke a role for protein diffusion, but proposes instead that a process of directed and regulated transport produces a concentration gradient of *bcd* mRNA that precedes and generates the Bcd protein gradient. The second model also involves a gradient of *bcd* RNA, but because the *bcd* RNA was not observed to extend as far posteriorly as Bcd protein, it proposes a major role for Bcd protein diffusion (*Little et al., 2011*). Although our studies of pre-syncytial blastoderm stages do not address the RNA and protein distributions at the later syncytial blastoderm stages, they are likely to be relevant to them.

We observed that *bcd* RNA and Bcd protein formed similar distributions in the interior of pre-syncytial blastoderm embryos (*Figures 4–7*). Although these distributions were constantly changing, the approximate match between the RNA and protein patterns remained constant. The RNA and protein were tightly concentrated at the anterior end of stage 14 oocytes, broader in nc1 embryos, and formed a graded internal plume that extended the farthest toward the posterior in nc3-4 embryos. The plume had complex geometries that were not radially symmetric, but after nc6-7, the internal plume was no longer visible and most of the *bcd* RNA and protein was near or at the cortex. The cortical distributions were radially symmetric. The mechanisms that generate these distributions are not known, but because the patterns of *bcd* RNA and protein are so similar, we suggest that the mechanisms are probably related. Although these mechanisms cannot be deduced from the geometries of the distributions, the complex and well-defined shapes and the rapidity with which the distributions form and change would seem to be incompatible with passive diffusion.

The *bcd* RNA and Bcd protein distributions we observed in the pre-cellular embryo reveal two distinct concentration gradients – one in the embryo interior that forms between nc1 and nc6, and a second one that forms later at the embryo cortex. We assume that the *bcd* RNA and Bcd protein of the early, first gradient contribute to the second, and therefore call this a two-step model. This model contrasts with the models that have been proposed previously in which the dispersion of protein (and RNA) from the anterior pole is continuous during both the pre-syncytial blastoderm and syncytial blastoderm stages (*Figure 9*). Although the two-step model is presumably generated by motor-driven directed movement, the high yolk content of the embryo cytoplasm has impeded analysis of its cytoskeletal elements. However, several studies have reported microtubule networks in the pre-syncytial embryo (*Fahmy et al., 2014*; *Karr and Alberts, 1986*) and threads of microtubules up to 50 µm long extending from the cortex into the interior at syncytial blastoderm stages

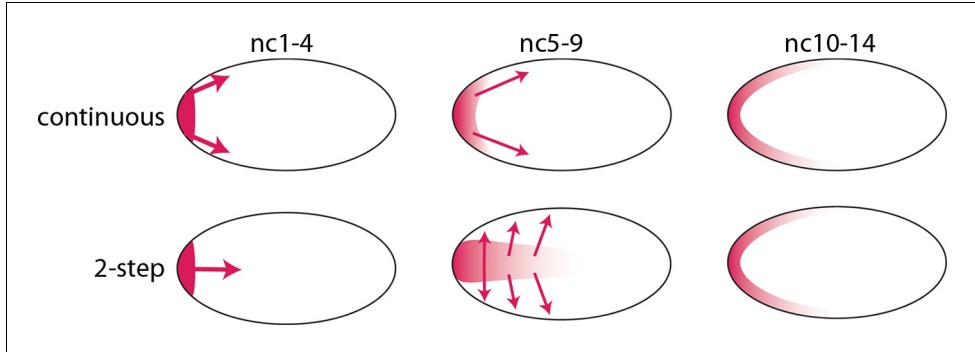

**Figure 9.** Models of Bcd gradient formation. Two contrasting models of Bcd gradient formation are depicted as viewed from a sagittal section in the middle of pre-cellular embryos, oriented anterior left. A model that assumes a continuous redistribution of *bcd* RNA and Bcd protein (red) from the anterior pole (upper row) contrasts with the two-step model (lower row) in which the *bcd* products generate a plume in the middle of the embryo during the first four nuclear cycles and then generate a second gradient at the cortex in the syncytial blastoderm stages (nc9-14).

(*Fahmy et al., 2014*). These networks offer a possible mechanism that might transport particles of *bcd* RNA (*Fahmy et al., 2014*; *Spirov et al., 2009*) and protein posteriorly to form the internal plume in pre-nc6 embryos (*Figures 4–7*), and to re-distribute *bcd* RNA and protein to the cortex at blastoderm stages.

## A role for Bicoid in the pre-syncytial blastoderm embryo

Our analysis of *Kr* expression revealed that transcription initiates as early as nc7 (*Figure 1*). Although the number of nuclei with nascent *Kr* transcripts was small, the internal nuclei with transcripts were at the approximate position along the anterior/posterior axis where older, syncytial blastoderm embryos have robust *Kr* expression, and this position was influenced by the level of Bcd protein in a similar way as the later, cortical expression. We also detected a concentration gradient of nuclear Bcd protein at the nc6 stage, when the embryo has just 32 nuclei that are deep in the interior (*Figure 5*), and observed that the both the internal and cortical Bcd gradients as well as both internal and cortical *Kr* expression were abnormal in *exu* and *stau* mutant embryos (*Figures 4* and *Figure 4—figure supplement 2*). These results show that Bcd regulates *Kr* in nuclei that are far from the cortex, and importantly, that the internal plume of Bcd protein is a functional distribution. These findings therefore establish that it is the pre-syncytial blastoderm, internal Bcd gradient that regulates gap gene expression and organizes the embryo's anterior/posterior axis. Although these findings do not imply what the role of the subsequent cortical Bcd gradient in the syncytial blastoderm stages Bcd gradient might be, it is most likely that the cortical gradient derives from the earlier distribution and therefore that it reflects the outputs of the pre-syncytial blastoderm, internal gradient.

## Perspectives

Morphogens such as Hedgehog, Wingless, Decapentaplegic, and Fibroblast growth factor distribute in concentration gradients across fields of cells in the tissues of developing animals. Their distributions are generated by transport along actin-based cytonemes (Roy, 2011 #56 and reviewed in *Kornberg, 2014*) and direct exchange between producing and receiving cells at morphogenetic synapses where release and uptake of secreted proteins is regulated (Roy, 2014 #57 and reviewed in *Kornberg and Roy, 2014*). The generation of the Bcd concentration gradients in the pre-cellular embryo would appear to have little in common with the gradients that form by cytoneme-mediated dispersion across fields of cells, but we pose the question whether they do. Neither appears to be dependent on passive diffusion and both appear to involve dispersion along cytoskeletal cables. The critical attribute that these mechanisms share is that they provide ways to regulate movement in space and time.

# Materials and methods

## Reagents

### Fly lines

(WT) *DGRP-208*; *stau$^1$ cn$^1$ bw$^1$/CyO*; *Df(2R)PC4/CyO*; *cn$^1$ exu$^4$ bw$^1$/CyO, l(2)DTS100$^1$*; *w\**; *Df(2R) exu$^2$/CyO*; *bcd$^{E1}$*; *Diap1$^1$ st$^1$ kni$^{ri-1}$ bcd$^6$ rn$^{roe-1}$ p$^p$/TM3, Sb$^1$*; *Dp(bcd:GFP)*; *bcd$^{E1}$/TM3*; *+/CyO, Dp (bcd)*; *bcd$^{E1}$/TM3*; *Df(2R)Kr$^{10}$, b$^1$ pr$^1$ Bl$^1$ c$^1$/SM1*; *Dp(bcd:GFP)* (4x *bcd*); *p(BAC, +5 +8)* (6x *bcd*)

### Antibodies

Primary antibodies: rabbit α-Bcd (M. Biggin); rabbit α-GFP (Life Technologies, Waltham, MA); sheep α-DIG (Roche Diagnostics, Pleasanton, CA); mouse α-Lamin (Developmental Studies Hybridoma Bank, Iowa City, IA). Secondary antibodies (Life Technologies): goat α-rabbit, Alexa 488; goat α-rabbit, Alexa 555; donkey α-rabbit, Alexa 555; donkey α-sheep, Alexa 488; donkey α-sheep, Alexa 555; goat α-mouse, Alexa 488; goat α-mouse, Alexa 555.

## Embryo in situ hybridization (modified from *Boettiger and Levine, 2013*)

### Fixation and de-vitellinization

Embryos were de-chorionated in 50% bleach, washed with 0.1% Triton X-100 and transferred to a 20 ml vial containing fixative (5 ml heptane, 5 ml phosphate buffered saline (PBS), 50 mM EGTA (pH 8.0), 10.24% formaldehyde (Ultrapure), and shaken for 1.5 hr. The lower phase was removed and embryos were de-vitellinized by vigorous shaking in 4 ml cold methanol (MeOH) for 1 min, transferred to a 1.5 ml (Eppendorf) tube, rinsed 3x with MeOH and stored at -20°C in MeOH.

### Dehydration

Embryos (0.05 ml) were rocked in 1:1 MeOH:EtOH in a 1.5 ml (Eppendorf) tube for 5 min, and rinsed with EtOH (2 x 5 min). 0.9 ml EtOH was removed, 0.9 ml xylene was added, and after rocking for 1 hr, liquid was removed and embryos were rinsed with 1 ml EtOH (2 x 5 min).

### Post-fixation

Embryos were rinsed by rocking in 1 ml MeOH (2 x 5 min), rinsed in 1:1 MeOH:0.5% formaldehyde in PBT (PBS, 0.1% Tween 20, 0.3% Triton X-100) for 5 min, rocked in 5% formaldehyde in PBT (1 hr), rinsed by rocking in 1 ml PBT (2x) followed by rocking in 1 ml PBT (4 x 10 min).

### Pre-hybridization

Embryos were incubated with rocking for: (1) 10 min at room temperature in 1 ml 1:1 hybridization solution (HybS; 50% formamide, 5xSSC, 100 μg/ml salmon sperm DNA, 50 μg/ml heparin, 0.1% Tween 20, 0.3% Triton X-100); (2) 10 min in 1 ml HybS at 55°C; (3) 45 min in 1 ml fresh HybS at 55°C; and (4) overnight in fresh HybS at 55°C.

### Hybridization

5 μl probe (prepared according to *Boettiger and Levine, 2013*) was diluted to 100 ng/μl, added to 45 μl HybS, incubated for 2 min at 82°C and 2 min on ice, and added to embryos that were drained of HybS, and incubated at 55°C for 24 hr.

### Staining

Embryos were rinsed at 55°C in 1 ml HybS (3x) that was pre-warmed to 55°C for 15 min, washed in 1 ml HybS (6 x 30 min), washed in 1:1 HybS:PBT at room temperature for 10 min and washed in PBT at room temperature for 30 min. Embryos were rocked in 1 ml blocking solution (1:5 Roche blocking reagent:PBT) for 1 hr, in 0.5 ml fresh blocking solution with 0.12 mg pre-adsorbed α-Digoxigenin antibody overnight at 4°C, washed in 1 ml PBT at room temperature (10 x 30 min) and in 1 ml blocking solution at room temperature for 1 hr. Embryos were incubated for 1.5 hr in 0.5 ml secondary antibody in blocking solution, rinsed in 1 ml PBT (2x), stained with DAPI (0.1 μg/ml) in 1 ml PBT

(5 min), washed in 1 ml PBT (3-4x) at 4°C (2–14 hr), mounted in Vectashield on a slide and sealed under a coverslip with nail polish.

## Embryo antibody staining (modified from *Boettiger and Levine, 2013*)

### Fixation and de-vitellinization

Embryos were de-chorionated in 50% bleach, washed with 1% Triton X-100 and transferred to a 20 ml vial containing fixative (4 ml heptane, 3 ml phosphate-buffered saline (PBS), 1 ml 16% formaldehyde (Ultrapure)) and shaken for 25 min The lower phase was removed and embryos were de-vitellinized by vigorous shaking in 4 ml cold MeOH for 1 min, transferred to a 1.5 ml (Eppendorf) tube, rinsed 3x with MeOH and stored in MeOH at -20°C.

### Prewash

Embryos were rinsed 2x with PBT (PBS, 0.1% Tween 20, 0.3% Triton X-100), washed by rocking for 4 x 15 min in 1 ml PBT, rocked for 1 hr in 1 ml PBTi (PBT, 250 mM imidazole), rocked for 2 hr in 1 ml hybridization buffer, rinsed in 1 ml PBT (2x) and rocked in 1 ml PBT for 15 min (Note: imidazole is included to block binding of DAPI to inorganic polyphosphate that is distributed in small granules throughout the embryo.)

### Staining

Embryos were incubated for 1 hr in 1 ml blocking solution followed by incubation for 14+ hr at 4°C in 0.5 ml blocking solution + primary antibody, rinsed by rocking at room temperature in 1 ml PBT for 30 min (10x), and blocked for 1 hr in 1 ml blocking solution. For secondary antibody, embryos were incubated with secondary antibody in 0.5 ml blocking solution for 1.5 hr at room temperature, rinsed in 1 ml PBT (2x), stained with DAPI in 1 ml PBT (5 min), washed in 1 ml PBT (3-4x) at 4°C for 2–14 hr, mounted in Vectashield on a slide and sealed under a coverslip with nail polish.

## Oocyte preparation

Stage 14 oocytes were dissected from 5 pairs of ovaries (including the complete female reproductive system) that were obtained from 4–5 day old, well-fed females. Stage 14a and 14b accounted for 96–97% of stage 14 oocytes. Approximately 3% were stage 14c and 14d. To obtain larger numbers of stage 14c and 14d oocytes, five pairs of ovaries were isolated and incubated in PBS for 20 hr at room temperature; the relative proportion of stage 14 oocytes in these preparations was 14a, 0%; 14b, 2–3%; 14c, 75–80%; 14d, 15–20%. Stage 14a and 14b oocytes were not extruded from the distal tip of the ovary; 14c and 14d oocytes were wholly or partially in a lateral branch of the oviduct. The stage 14c and 14d oocytes were sensitive to hypochlorite treatment and had not completed meiosis I, and were therefore designated as not activated (*Sartain and Wolfner, 2013*). Oocytes isolated either without or with incubation were subdivided and either stained with DAPI or processed for in situ hybridization or antibody staining. Results of in situ hybridization and antibody staining with oocytes obtained from freshly dissected ovaries and from ovaries incubated ex vivo were indistinguishable.

## Stage 14 oocyte antibody staining (modified from *Boettiger and Levine, 2013*)

### Fixation

Ovaries were dissected in PBS from 4–5 day old females. Stage 14 oocytes were separated from the ovaries and transferred to a 20 ml vial containing fixative (5 ml heptane, 5 ml phosphate buffered saline (PBS), 50 mM EGTA (pH 8.0), 10.24% formaldehyde (Ultrapure), and shaken for 1 hr. The lower phase was removed, 5 ml cold MeOH was added and after 1 min vigorous shaking, the upper phase was removed and oocytes were transferred to a 1.5 ml (Eppendorf) tube, rinsed with MeOH (3x) and stored at -20°C in MeOH.

### De-chorionation and de-vitellinization

Fixed oocytes in MeOH were transferred to a glass cavity slide and viewed under a dissecting microscope. Fine forceps were used to hold the dorsal appendages and the dorso-lateral side of the

chorion and vitelline membrane were sliced with a fine needle. Oocytes freed of chorion and vitelline membrane were stored at -20°C in MeOH.

### Pre-wash
Oocytes were rinsed 2x with 1 ml PBT (PBS, 0.1% Tween 20, 0.3% Triton X-100) and with rocking in 1 ml PBT (4 x 15 min, were incubated in 1 ml PBTi (PBT, 250 mM imadozole) for 1 hr, in 1 ml HybS for 2 hr, and rinsed with 1 ml PBT (2x) and with rocking in 1 ml PBT for 15 min.

### Staining
Oocytes were rocked in 1 ml blocking solution (1:5 Roche blocking reagent:PBT) for 1 hr, and in 0.5 ml pre-adsorbed primary antibody in blocking solution at 4°C overnight, washed with rocking in 1 ml PBT at room temperature (10 x 30 min, and 1 ml blocking solution for 1 hr. Oocytes were incubated with secondary antibody in 0.5 ml blocking solution for 1.5 hr at room temperature, rinsed with 1 ml PBT, and stained with DAPI in 1 ml PBT for 5 min, washed in 1 ml PBT (3-4x) at 4°C for 2–14 hr, mounted in Vectashield on a slide and sealed under a coverslip with nail polish.

## Stage 14 oocyte in situ hybridization
### Fixation, de-chorionation and de-vitellinization
Same as for oocyte antibody staining (above).

### Dehydration, post-fixation, pre-hybridization, hybridization, wash and antibody staining
Same as for embryo in situ hybridization (above).

## Histology
Embryos were mounted with slight pressure under coverslips and orientation was determined by analysis of serial optical sections. Images were obtained with a Leica SPE confocal microscope and processed with ImageJ.

## Q-PCR
Total RNA was prepared from five embryos with Zymo Research RNA MicroPrep kits (Cat. #R1060) and quantified by absorbance with a nanodrop spectrophotometer. cDNA was prepared using Applied Biosystem High Capacity RNA-to-cDNA kits (Cat. #4387406) starting with approximately 200 ng RNA. The Q-PCR reactions were carried out with a BioRad C1000 Touch Thermal Cycler and SsoAdvanced SYBR Green according to manufacturers' instructions.

## Acknowledgements
We thank: Drs. G Struhl, T Gregor, E Gavis, S Small, and the Bloomington Stock Center for fly stocks; M Biggin for α-Bcd antibody and the Developmental Studies Hybridoma Bank for α-lamin antibody; and P Rao, S Small, M Noll, M Wolfner and Zac Kornberg for discussions and constructive suggestions. This work was funded in part by the UCSF Program for Breakthrough Biomedical Research and the Sandler Foundation.

## Additional information

### Funding

| Funder | Grant reference number | Author |
| --- | --- | --- |
| Sandler Foundation | | Thomas B Kornberg |
| National Institutes of Health | GM109410 | Thomas B Kornberg |

The funders had no role in study design, data collection and interpretation, or the decision to submit the work for publication.

## Author contributions

ZA-M, Conception and design, Acquisition of data, Analysis and interpretation of data; TBK, Conception and design, Analysis and interpretation of data, Drafting or revising the article

## Author ORCIDs

Thomas B Kornberg, http://orcid.org/0000-0002-6879-7066

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
