## [Decision Letter]

Thank you for submitting your work entitled "Bicoid gradient formation and function in the *Drosophila* pre-syncytial blastoderm" for consideration by *eLife*. Your article has been reviewed by three peer reviewers, and the evaluation has been overseen by Allan Spradling as the Reviewing Editor and James Manley as the Senior Editor.

The following individuals involved in review of your submission have agreed to reveal their identity: Michael Eisen and Markus Noll (peer reviewers).

The reviewers have discussed the reviews with one another and the Reviewing Editor has drafted this decision to help you prepare a revised submission.

The *Drosophila* egg and early embryo provide much of our current insight into how maternal factors contribute to early developmental patterning. The formation of the bicoid protein gradient by the regulated translation of maternally localized *bcd* mRNA represents a key aspect of this process, one that has become the textbook example of a "morphogen gradient." However, the mechanisms which shape the gradient remain imperfectly understood and competing models have been put forward. The consensus opinion among the reviewers is that the work presented here adds to our understanding of *Drosophila* embryo patterning and Bcd gradient formation.

Several important findings are reported. (1) The gap gene *Kr* is expressed as nascent transcripts by nuclear cycle 7 and 8 (nc7 and 8) at the proper location (2) in nuclei that are already polarized (as these transcripts are associated with chromosome arms and located basally; *Kr* is located at the telomere), i.e., prior to their arrival at the cortex. (3) Bcd protein is visible in a concentration gradient already at nc2 to nc6, which depends also at early presyncytial blastoderm stages on the presence of maternal Stau and Exu proteins. (4) At these early presyncytial blastoderm stages, the Bcd protein gradient extends posteriorly in what the authors call a medial plume rather than at the cortex as at later stages. This proves that the process of gradient formation must be more complex than previously thought. (5) The distribution of *bcd* RNA is very similar to that of Bcd protein also at these early cleavage stages (nc2 – nc6), with the medial plume reaching its maximum intensity at nc4. (6) The authors detected Bcd protein already at nc1 and (7), most astonishingly, in stage 14c and 14d oocytes, again coincident with the anterior location of *bcd* RNA in oocytes. (8) In contrast to *bcd* RNA, which was evident in oocytes at earlier stages, Bcd protein was not detected before stage 14c. This demonstrates that the translational block of *bcd* mRNA is relieved prior to fertilization and that Bcd protein is present precisely where *bcd* mRNA is sequestered.

The authors describe a different shape and demonstrate that zygotic gene expression starts earlier than the pre-cellular blastoderm stage by forcing a revision in the previously accepted timing and spatial distribution of *bcd* RNA and protein, and by contradicting some aspects of the leading model for how the gradient is produced.

In the Discussion, the authors propose a two-step model of Bcd gradient formation, which differs from an earlier model mainly by the initial gradient formation of the medial plume during nc2-6. This gradient is followed by a cortical gradient, consistent with a model previously proposed (Spirov et al., 2009). This latter model and the model proposed by the authors are consistent with the gradient being formed by the *bcd* mRNA that, by translation, produces the Bcd protein gradient although the authors are more conservative. The most important conclusion is that mechanisms of morphogen gradient formation in general are not dependent on passive diffusion, overthrowing a tenet maintained over many decades, but "appear to involve dispersion along cytoskeletal cables."

A small number of relatively minor revisions are now requested.

1) The major weakness of the paper relates to the authors' claim to have developed new and more sensitive methods for detecting RNA and protein in preblastoderm *Drosophila* embryos. While the authors document expression earlier than previously reported, an achievement that likely required improved sensitivity of DIG in situ hybridization as well as of antibody staining of early embryos to achieve a much better signal-to-noise ratio. However, the manuscript does not include experiments directly comparing different in situ hybridization methods, including in particular smFISH, which is generally accepted to have greater sensitivity in many applications. It remains unclear why the low level expression described here was not detected in previous studies. Consequently, statements about the relative sensitivity of various methods should be removed from the manuscript unless data from such tests are added.

2) The finding that the *bcd* RNA translational block is relieved in late stage 14 oocytes is interesting but because follicles were incubated in vitro for an extended period, the possibility of artifact remains. A recent study shows that stage 14 oocytes about to be ovulated can be recognized by the loss of posterior follicle cells (Deady et al. 2015). The authors may wish to investigate whether the onset of *bcd* translation takes place before or after ovulation, in vivo.

3) The authors should more adequately reference earlier literature, which shows that nuclear foci of in situ hybridization are often sites of nascent transcription.

---

## [Author Response]

*A small number of relatively minor revisions are now requested. 1) The major weakness of the paper relates to the authors' claim to have developed new and more sensitive methods for detecting RNA and protein in preblastoderm Drosophila embryos. While the authors document expression earlier than previously reported, an achievement that likely required improved sensitivity of DIG in situ hybridization as well as of antibody staining of early embryos to achieve a much better signal-to-noise ratio. However, the manuscript does not include experiments directly comparing different in situ hybridization methods, including in particular smFISH, which is generally accepted to have greater sensitivity in many applications. It remains unclear why the low level expression described here was not detected in previous studies. Consequently, statements about the relative sensitivity of various methods should be removed from the manuscript unless data from such tests are added.*

A supplemental figure to Figure 1 has been added that compares results of *bcd* and *Kr* in situ hybridization with FISH, Quantigene and DIG probes. The text in the Results now reads:

“To identify the nuclei that express *Kr* transcription prior to nc10, we evaluated various different techniques for in situ hybridization, and determined that a procedure we developed that uses DIG probes was the most sensitive for our studies of *Kr* and *bcd* transcripts in pre-cellular embryos (see Materials and methods and Figure 1—figure supplement 1).”

*2) The finding that the bcd RNA translational block is relieved in late stage 14 oocytes is interesting but because follicles were incubated in vitro for an extended period, the possibility of artifact remains. A recent study shows that stage 14 oocytes about to be ovulated can be recognized by the loss of posterior follicle cells (Deady et al. 2015). The authors may wish to investigate whether the onset of bcd translation takes place before or after ovulation, in vivo.*

The following sentence has been added to the Methods:

“Results of in situ hybridization and antibody staining with oocytes obtained from freshly dissected ovaries and from ovaries incubated ex vivo were indistinguishable.”

*3) The authors should more adequately reference earlier literature, which shows that nuclear foci of in situ hybridization are often sites of nascent transcription.*

The earlier literature showing foci of in situ hybridization as sites of nascent transcript has been cited as follows:

“In situ hybridization can detect sites of nascent transcript production as points of staining or fluorescence (Femino et al., 1998; Shermoen and O'Farrell, 1991), and the nc10-13 embryos we analyzed had bright dots in most or all nuclei in the “*Kr* band”.”